# Effect of ZrB_2_ Content on the Properties of Copper Matrix Composite

**DOI:** 10.3390/ma17246105

**Published:** 2024-12-13

**Authors:** Iwona Sulima, Remigiusz Kowalik, Michał Stępień, Paweł Hyjek

**Affiliations:** 1Institute of Technology, University of the National Education Commission, Krakow, Podchorazych 2 Str., 30-084 Krakow, Poland; iwona.sulima@uken.krakow.pl (I.S.); pawel.hyjek@uken.krakow.pl (P.H.); 2Faculty of Non-Ferrous Metals, AGH University of Krakow, Mickiewicz 30 Av., 30-059 Krakow, Poland; mstepien@agh.edu.pl

**Keywords:** copper matrix composites, ZrB_2_, powder metallurgy, mechanical properties, wear resistance, corrosion resistance

## Abstract

This study examined the physical, mechanical, tribological, and corrosion properties of copper metal matrix composites reinforced with zirconium diboride (ZrB_2_). Cu-xZrB_2_ composites (x = 0.5, 10, 15, 20 wt.%) were produced by the ball-milling process and spark plasma sintering (SPS). Introducing ZrB_2_ particles into copper matrix composites significantly improves their mechanical and tribological properties while deteriorating their density, porosity, and corrosion properties. It was shown that the relative densities of the composites gradually decreased from 96% to 90%, with an increase in the ZrB_2_ content to 20 wt.%. Likewise, hardness, compressive strength, and wear resistance improved with increasing ZrB_2_ content in the copper matrix. Corrosion resistance tests in a 0.05 M sulfuric acid environment showed a disproportionate decrease in the resistance of this composite with an increase in the concentration of the ceramic phase compared to other environments.

## 1. Introduction

For modern technology and industry needs, continuous development of new materials with unique properties is required. Their production by conventional methods is often very difficult or impossible. Such materials include, among others, metal matrix composites, which are a very dynamically developing group of materials [1]. The most commonly used matrix materials of composites are magnesium [2], aluminum [3], titanium [4], iron [5], nickel [6], and copper [7]. Typical reinforcing phases are carbides (TiC, SiC) [8,9], oxides (ZrO_2_, Al_2_O_3_, TiO_2_, SiO_2_) [10,11,12], nitrides (Si_3_N_4_, BN) [13,14] and borides (TiB_2_, ZrB_2_) [15,16]. In recent years, there has been a new trend of using attractive ceramic materials, such as borides, as reinforcement for metal composites [15,17,18,19]. Among these materials, zirconium diboride is often chosen due to its favorable combination of properties, which include a high melting point, low density, high modulus of elasticity, excellent hardness, electrical and thermal stability, as well as excellent wear resistance and good corrosion resistance [20,21,22]. Copper matrix composites are materials whose use requires excellent mechanical properties as well as high electrical or thermal conductivity. Mechanical properties can be effectively improved by adding a ceramic reinforcing phase to the copper matrix. However, this often affects the deterioration of the conductivity of the composites. Therefore, a crucial aspect in the design of copper matrix composites is to produce a material with good mechanical properties and good conductivity [23,24].

Copper matrix composites can be effectively manufactured by various powder metallurgy techniques, including conventional sintering [25], hot pressing (HP) [26], hot isostatic pressing (HIP) [27], and Spark Plasma Sintering (SPS) [28,29]. Table 1 presents selected data from the literature [23,30,31,32,33] regarding the influence of the type and amount of ceramic phases on the density and hardness of copper matrix composites.

SPS is an advanced manufacturing technology that enables the production of very dense composite materials. The use of this modern sintering technology allows for the reduction of the sintering temperature, shortens the sintering time, and allows the sintering process to be conducted without the use of activators. This positively impacts the economic aspect of manufactured composite materials [30,34]. In the literature [31,32,33,35,36], several studies have focused on producing copper matrix composites using the SPS method. Long et al. [32] sintered Cu–NbC composites using spark plasma sintering temperatures between 900 °C and 1000 °C for 15 min. The results showed that the composites’ density, hardness, and electrical conductivity proved to be a function of temperature. Simultaneously, it was observed that the hardness increased with the increase in the NbC phase while the electrical conductivity deteriorated. In turn, Wang et al. [31] investigated the effect of various SPS parameters, temperature (600–900 °C), and pressure (10–80 MPa) on the electrical and mechanical properties of Cu-TiC composites. It was shown that the increase in sintering temperature significantly improves the density, microhardness, and electrical conductivity. Analyzing the results, it was noticed that density, microhardness, and electrical conductivity showed a positive correlation with the sintering pressure. Soloviova et al. produced Cu-(LaB_6_-TiB_2_) composites using the SPS method. They obtained a twofold increase in the density of the material as a function of the sintering temperature. A positive effect of the temperature change from 850 °C to 1050 °C of sintering on the hardness and electrical conductivity of the composites was demonstrated.

**Table 1 materials-17-06105-t001:** The influence of ceramic particles on density and hardness of the Cu-based composites.

Cu-Based Composites	Sintering Condition	Density	Hardness	Ref.
Cu-5 vol.% TiC	SPS; 800 °C; 10–80 MPa; 5 min	7.0–8.6 g/cm^3^	125–268 HV1	[31]
Cu-1 vol.% NbC	SPS; 1000 °C; 100 MPa; 10 min	98.1%	162 HV	[32]
Cu-5 vol.% NbC		97.0%	268 HV	
Cu-15 vol.% NbC		96.8%	395 HV	
Cu-25 vol.% NbC		97.0%	452 HV	
Cu-1 vol.% Al_2_O_3_	SPS; 700 °C; 10–50 MPa; 5 min	93.2%	77 HV0.3	[33]
Cu-5 vol.% Al_2_O_3_		92.8%	125 HV0.3	
Cu-7 vol.% Al_2_O_3_		86.1%	75 HV0.3	
Cu-1 wt.% ZrB_2_	Hot-pressed sintering; 840 °C; 25 MPa; 2 h	96%	69 HV0.2	[37]
Cu-3 wt.% ZrB_2_		95%	84 HV0.2	
Cu-5 wt.% ZrB_2_		92.5%	93 HV0.2	
Cu-7 wt.% ZrB_2_		91.8%	100.8 HV0.2	
Cu-9 wt.% ZrB_2_		91.3%	82 HV0.2	
Cu-2 wt.% Mo_2_C	Hot-pressed sintering; 880 °C; 20 MPa; 10 min	91.5%	58.9 HV	[38]
Cu-5 wt.% Mo_2_C		91.2%	65.8 HV	
Cu-7 wt.% Mo_2_C		91.3%	69.6 HV	
Cu-1wt.% Y_2_O_3_	SPS; 900 °C; 50 MPa; 5 min	93.9%	101.3 HV0.1	[39]
Cu-3 wt.% Y_2_O_3_		100%	125.7 HV0.1	
Cu-5 wt.% Y_2_O_3_		100%	140.5 HV0.1	

In this paper, Cu-ZrB_2_ composites were prepared using the Spark Plasma Sintering (SPS) method. The influence of different zirconium diboride content (5, 10, 15, and 20 wt.%) on the properties of sintered materials, such as density, portability, Young’s modulus, hardness, compressive strength, corrosion resistance, and abrasive wear resistance, was analyzed. The results will help to understand the importance and influence of the amount of zirconium diboride in optimizing the properties of such composite materials.

## 2. Materials and Methods

This study used commercially available Cu powder (10 μm, 99.9 wt.% purity, Kamb Import Export, Warsaw, Poland) and ZrB2 powder (2.5–5.5 μm, 99.9 wt.%, H.C. Starck Tungsten GmbH, Goslar, Germany) to fabricate composites. The microstructures of raw powders are shown in Figure 1 and Figure 2, which correspond to the results of the particle size tests. The morphology of the starting powders was evaluated using scanning electron microscopy (JEOL JSM 6610LV, Tokyo, Japan). Measurements of the particle size distribution of the starting powders and composite mixtures were performed in polypropylene alcohol using the SALD-7500 nano analyzer (Shimadzu Corporation, Kyoto, Japan) with Wing SALD II software (Version 3.4.9, Shimadzu Corporation, Kyoto, Japan). Measurements were carried out with a measurement step of 1 s.

Next, the mixture powders were milled using a Fritsch Pulveristte 5 planetary ball mill (Fritsch GmbH, Idar-Oberstein, Germany) with a ball-to-powder weight ratio of 5:1. The container and balls were made of tungsten carbide. Milling was carried out at a speed of 200 rpm for 10 h with a repeating cycle: 20 min of milling—10 min break. Four powder mixtures were prepared, which contained the following amounts of the ZrB_2_: 5 wt.%, 10 wt.%, 15% wt.%, and 20 wt.%. Figure 3 shows the powder mixture’s morphology and particle size distribution containing 20 wt.% ZrB_2_ after 10 h of milling.

The LSP 100 laboratory sinter press (Dr. Frisch, Fellbach, Germany) was used for sintering the copper and Cu-ZrB_2_ composites. The powders were sintered at 35 MPa in an argon atmosphere at 900 °C for 3 min. The furnace heating rate was kept at 100 °C/min. Figure 4 shows the heating curve registered during the SPS process.

The sintered samples were characterized by X-ray diffraction (XRD) to identify the phases. The studies were performed with a Rigaku Miniflex II X-ray diffractometer using Cu-K radiation (Tokyo, Japan). Microstructure characterization was carried out using the OLYMPUS LEXT OLS5100 confocal microscope (Olympus, Tokyo, Japan) and scanning electron microscopy (SEM, JEOL JSM 6610LV, Tokyo, Japan) equipped with energy-dispersive spectroscopy (EDS, Aztec, Oxford Instruments, Oxford, UK).

Hardness was measured using NEXUS 4000 Vicker’s microhardness tester (Innovatest, EuropeBV, Maastricht, The Netherlands) with a load of 2.942 N. A minimum of ten measurements on each sample were implemented for hardness evaluations. Compression tests were conducted on an INSTRON TT-DM testing machine (Norwood, MA, USA) with a crosshead speed of 1 × 10^−4^ mm/s. The tests were carried out at room temperature in accordance with the requirements of the ISO standard [40].

The corrosion test was carried out using a 0.05 M H_2_SO_4_ solution. All measurements were conducted at 25 ± 1 °C in a deaerated solution. The solution was deaerated by bubbling argon (175 mL/min) 30 min before sample mounting, and argon flow continued throughout the experiment. The exanimated surface was polished on a P800 grid of abrasive paper, cleaned in an ultrasonic bath in ethanol, and dried in a hot air stream (60 °C). A conventional three-electrode system was applied with Ag/AgCl (3M KCl) as the reference electrode and the platinum wire as a counter electrode. The experiments were performed on the potentiostat of AUTOLAB PGSTAT128 (Metrohm AG, Herisau, Switzerland). Before each polarization test, a sample was conditioned for 30 min in the solution to obtain an open current potential (OCP) value. The potentiodynamic polarization scan was performed from −0.20 V vs. OCP in an anodic direction at a scan rate of 1 mV/s. The direction was changed when the current density reached 5 mA/cm^2,^ and the scan was ended at the initial potential value. The measurements were repeated three times to obtain more reliable results, and the measurement uncertainty was given as standard deviation.

The wear behavior of the sintered materials was assessed using a ball-on-disc tribometer (Elbit Company, Koszyce Małe, Poland) as per the ISO standard [41]. Wear tests were conducted in a dry environment. Wear experiments were carried out under the following conditions: time of 10,000 s, load of 5 N, distance of 1000 m, and rotational speed of 200 rpm. A counter-sample made of Al_2_O_3_ (diameter of 3.175 mm) was used. Before each test, the samples and the steel balls were cleaned with acetone in an ultrasonic washer. The friction coefficient was measured continuously during the test following the established wear conditions. Mass loss (Δm) was determined using the weight loss method. The samples were weighed before and after each wear test using an analytical balance RADWAG AS 220/C/2 (Radwag, Radom, Poland) with a precision of 0.1 mg. Next, the mass loss was calculated.

In the next step, the wear track was observed using an OLYMPUS LEXT OLS5100 confocal microscope and scanning electron microscopy (SEM) using Jeol JSM 6610 LV. The average width of the wear path was determined based on 50 measurements taken for each analyzed wear. Figure 5 shows a part of the wear track marked with sample width measurements. The wear volume of the disc was determined, and then the wear coefficient was calculated according to the formula [41]:(1)$$W_{V(disc)}=\frac{V_{disc}}{F_{n}*L}$$
where:

F_n_—applied load [N]

V_disc_—wear volume of disc specimen [mm^3^]

L—sliding distance [m].

**Figure 5 materials-17-06105-f005:**
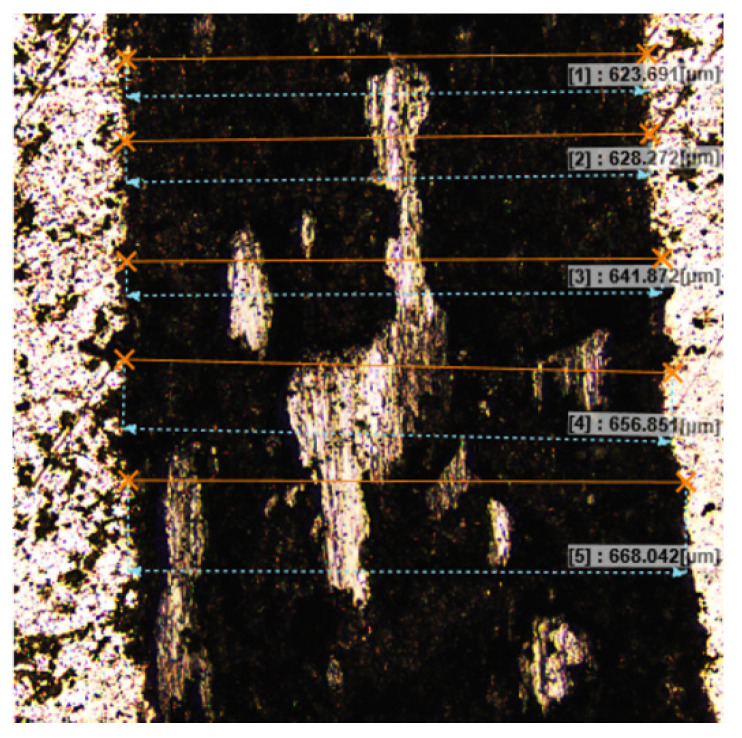
Example measurements of the average width of wear track for Cu-5%ZrB_2_ composite.

## 3. Results and Discussion

### 3.1. Physical and Mechanical Properties and Microstructure of the Sintered Materials

Table 2 shows the changes in measured density, porosity, and Young’s modulus of sintered copper and composites. It was shown that the relative density of Cu-xZrB_2_ composites (x = 0, 5, 10, 15, 20 wt.%) decreased with the increasing amount of ZrB_2_ particles, which were 98%, 96%, 94%, 91%, and 90%, respectively. Simultaneously, porosity increased with the increase in the reinforcing phase. Similar results were reported for other ceramic phase-reinforced copper matrix composites. Shaik and Golla [42] found that the relative density of copper matrix composites decreased from 99.76% for pure copper to 96.02% for the Cu-10 wt.% ZB_2_ composite. Zhang et al. [43] showed that in Cu-YB_2_C_2_ composites, the relative density decreased from 99.86% for pure copper to 94.15% for the composite containing 20 wt.% YB_2_C_2_.

It was observed that the values of Young’s modulus and hardness depended on the amount of ZrB_2_ in the matrix. Young’s modulus gradually increased with the amount of ZrB_2_ in the copper matrix, from 107 GPa for pure copper to 148 GPa for Cu-20% ZrB_2_ composites. The hardness measurement results indicated that adding 5 wt.% of zirconium diboride nearly doubled the hardness compared to copper (Cu-0% ZrB_2_). Fine hard ceramic particles ZrB_2_ in the copper matrix can act as obstacles during the pressing of the indenter, contributing to the increase in the hardness of the prepared composite. Consequently, increasing the content of the reinforcing phase provides more excellent resistance to plastic deformation, thus increasing the hardness of the composite. Therefore, the composites containing 20 wt.% ZrB_2_ showed three times higher hardness than sintered copper (Cu-0% ZrB_2_). The hardness of copper and the Cu-20% ZrB_2_ composite was 61.2 HV0.3 and 161 HV0.3, respectively.

Figure 6 shows the summary results of compressive strength tests of sintered materials. Similar characteristics of the true stress–strain curves were obtained for all composites with different plastic strain ranges. It was observed that, after reaching the maximum true stress, the stress value gradually decreased for each tested composite material. For comparison, sintered copper samples were deformed plastically during the tests and did not crack or break. Therefore, copper samples were deformed to the appropriate level of deformation. It was observed that the presence of hard ZrB_2_ particles in the copper matrix affects the mechanical properties of the composites. Based on the results of experimental studies obtained at room temperature, it was found that with the increase in the amount of ZrB_2_ phase in the matrix, the compressive strength of the tested materials improves, while the plastic properties deteriorate. The maximum true stress value was the highest for composites containing 20% ZrB_2_ and 364 MPa. Another effect was observed in the studies of Zhai et al. [44], where the reinforcing phase of the copper matrix was carbides (TiC). It was shown that the ductility and strength of the composite were the best when the mass fraction of TiC was 10%. For this composite, the maximum compressive strength was 650 MPa. Then, the compressive strain decreased with the increase of TiC content. When the mass fraction of TiC reached 25 wt.%, the composite’s yield strength and compressive strain decreased rapidly.

Figure 7, Figure 8 and Figure 9 show the comparative microstructures of sintered composites. EDS and X-ray techniques were used to verify the phase composition of the sintered composites. The reinforcing phase ZrB_2_ is distributed homogeneously throughout the volume of the copper matrix. The results of phase analyses confirmed the presence of the ZrB_2_ phase for all sintered composites (Figure 10). The identified WC phase (Figure 10) is a contamination resulting from the milling of composite powders in a tungsten carbide container.

### 3.2. Corrosion Behavior

In general, copper cannot oxidize in an acid solution with hydrogen displacement without dissolved oxygen. The reactions occurring during this process are presented in Equations (2)–(4), with the standard potentials vs. SHE (relative to Ag/AgCl 3 M KCl). The corrosion potential in a deaerated acid solution is determined by the two reactions: anodic copper dissolution (2) and cathodic hydrogen evolution (3).
(2)$$Cu-2e\to {Cu}^{2+}E^{0}=0.337\left(+0.127\right)[V]$$
(3)$$2H^{+}+2e^{-}\to H_{2}E^{0}=0.000\left(-0.210\right)[V]$$
(4)$$O_{2}+{4H}^{-}+4e^{-}\to 2H_{2}OE^{0}=1.230\left(+1.030\right)[V]$$

The slightly positive shift of open circuit potential (Figure 11) in a sintered Cu sample without ZrB_2_ compared to the solid copper sample may elucidate increased oxygen content on the surface of powder particles and trapped in the pores. According to potential mixed potential theory, without Reaction (4), the potential is shifted in the direction of Reactions (2) and (3). As time passes and the oxygen bound on the surface of the powder sample particles is depleted, the equilibrium potential tends to the potential of a solid pure copper sample. However, this mechanism cannot explain the significant shift in the open circuit potential towards the negative side of samples by adding the ZrB_2_ ceramic phase. In a similar acid sulfate solution (0.25 M Na_2_SO_4_ + 0.05 M H_2_SO_4_, pH = 1), pure ZrB_2_ ceramics show corrosion potential of about −0.4 V vs. Ag/AgCl electrode and expected corrosion current on 3 μA/cm^2^ [45]. So, it is possible, based on the mixed potential theory, that OCP (also corrosion potential obtained from Tafel extrapolation) is shifted by the presence of the ceramic phase, not by the effect caused by the porosity of the samples.

Polarization curves collected during the experiment are presented in Figure 12. On their basis, the corrosion potential and current, as well as the cathodic and anodic Tafel constants, were calculated using Tafel extrapolation, and the results are gathered in Table 3. The solid Cu sample and the sintered Cu sample without adding ZrB_2_ do not differ significantly, apart from the difference in the slope of the cathode part, which is probably related to diffusion. The polarization curves show a significant difference in the corrosion behavior of samples with and without adding the ZrB_2_ ceramic phase. The corrosion potential of samples with ZrB_2_ shifted about 300 mV toward the less noble direction (according to the OCP value). The corrosion current also increases strongly with the increase in the content of the ZrB_2_ phase. This increase is disproportionately large compared to similar samples tested in chloride solutions (3.5 wt.% NaCl) with a pH close to natural [46]. In a chloride environment, the corrosion current increased approximately 40 times in the worst possible case, while in a sulfate environment, it increased approximately 300 times.

Moreover, the increase in the corrosion current for ZrB_2_ in a sulfate environment (0.1 M H_2_SO_4_) in a 316L stainless steel matrix is also approximately two times lower than in this case [5]. The corrosion process of ZrB_2_ in acid solutions is most likely responsible for such a sharp increase in corrosion current, where ZrB_2_ is decomposed according to Reactions (4) proposed by Monticelli [42]
(5)$${ZrB}_{2}+{8H}_{2}O\to {ZrO}_{2}↓+2H_{3}{BO}_{4}+10H^{+}+10e^{-}$$

This process produces boric acid, which further acidifies the corrosion environment, enhancing the effect of crevice corrosion in the pores. In the case of a stainless steel matrix, under favorable conditions, matrix corrosion can be slowed by passivation. However, the copper matrix needs to provide such a chance. Moreover, the stability of the oxide layer formed on ZrB_2_ particles, which slows down the corrosion process, depends on the type of soluble complex and the pH value. In the case of acid sulfate solution (reaction 6), the equilibrium is shifted towards the right side due to higher complex stability [45,47].
(6)$${ZrO}_{2}↓+2{SO}_{4}^{2-}+2H^{+}↔{\left[ZrO{\left({SO}_{4}\right)}_{2}\right]}^{2-}+H_{2}O$$

This accelerates corrosion in sulfate solutions, but this phenomenon is inhibited in chloride solutions due to the low stability of the [ZrCl_6_]^2−^ complex. Moreover, the lack of oxygen dissolved in water additionally inhibits the formation of ZrO_2_, which also accelerates the corrosion process of ZrB_2_.

On the other hand, the presence of oxygen would accelerate the corrosion of the copper matrix. Therefore, the selected test conditions are extremely unfavorable from the point of view of the tested material. However, due to its good mechanical properties, it is worth selecting appropriate corrosion inhibitors in subsequent work.

### 3.3. Wear Behavior

The results of the friction coefficient as a function of time are shown in Figure 13. It was found that the friction coefficient of copper matrix composites decreased significantly when a higher ZrB_2_ content was used. Comparatively, for pure copper (Cu-0% ZrB_2_) and the Cu-20% ZrB_2_ composite, the friction coefficient was 0.76 and 0.44, respectively. It was observed that the steady-state friction coefficient was achieved after about 4000 s for pure Cu, while for Cu-ZrB_2_ composites, the friction coefficient stabilized after about 2000 s. Similar relationships were observed in other studies [42,44]. Shaik et al. [42] studied Cu-ZrB_2_ composites produced using a hot pressing method using the pin-on-disc method. They showed that the friction coefficient of Cu decreased from 0.56 to 0.16 after adding 10% ZrB_2_. In their studies, Zhou et al. [48] introduced a reinforcing phase WS_2_ (20 wt.%) with particle sizes of 0.6 and 5.0 μm into the copper matrix, which allowed them to achieve the friction coefficient of 0.158 and 0.172, respectively.

The conducted tests (Figure 14) showed that copper (Cu-0% ZrB_2_) shows the largest mass loss and, as expected, is characterized by the highest specific wear rate (3.48 × 10^−6^ mm^3^/Nm). It can be seen that adding only 5 wt.% ZB_2_ causes a more than two-fold decrease in the specific wear rate value (1.28 × 10^−6^ mm^3^/Nm). Adding more of the reinforcing phase, ZrB_2_ causes a gradual reduction in the specific wear rate. For composites containing 20 wt.% ZrB_2_, the specific wear rate is 0.42 × 10^−6^ mm^3^/Nm. Based on the results, it was found that the wear resistance of copper-based composites increases with the increase of ZrB_2_ content. This is because the Al_2_O_3_ ball can easily penetrate the soft copper matrix during the ball-on-disc test, resulting in excessive material removal from the worn surface.

In contrast, material removal is more limited in sintered composites because the ZrB_2_ reinforcing phase acts as a barrier and protects the copper matrix from the Al_2_O_3_ ball. By analyzing the results of the mass loss measurement results, it can be concluded that its value decreases with the increase in the reinforcing phase ZrB_2_. This is because the addition of ceramic particles ZrB_2_ significantly increases the hardness of copper, as shown in Table 1. Similar relationships were demonstrated in other studies of copper matrix composites [42,43,44]. Fathy et al. [49] showed that pure copper exhibits an exceptionally high wear rate compared to Cu-ZrB_2_ nanocomposites. Their studies using the pin-on-disc method used different normal loads and sliding speeds. Generally, it was shown that the wear rate of nanocomposites decreases with the increase of ZrO_2_ content (0, 3, 6, and 9 wt.%). In turn, Tjong and Lau [50,51] showed that adding 5% SiC effectively affects the wear resistance of Cu-SiC composites. A further increase in SiC content resulted in a significant reduction in volume loss, especially in the case of the composite containing 20 vol.% SiC.

Figure 15 shows examples of wear path microstructures for sintered composites. Generally, for all Cu-xZrB_2_ composites (x = 5, 10, 15, 20 wt.%), a very similar material removal mechanism by the Al_2_O_3_ ball from the wear track was observed. In the area of abrasion, scratches and abrasive grooves are visible. Additionally, the worn surfaces of the composites were subject to plastic deformation, which resulted in delamination of the material. During the tests, the material was permanently deformed and gradually abraded to the edge of the wear track. This suggests a mixed nature of wear of the tested materials. Microstructural observations indicate abrasive and adhesive wear of the composite materials.

Figure 16 shows examples of wear tracks and their profiles measured for copper and composites with different ZrB_2_ contents. 3D analyses of the profiles showed differences in the dimensions of the wear trace. A decrease in the width of the wear track was observed with the increase in the amount of the reinforcing phase (Figure 17). These observations confirmed the results of wear measurements of the sintered composite samples. As mentioned earlier, adding more ZrB_2_ to the copper matrix causes a gradual decrease in the specific wear rate. The hard ZrB_2_ reinforcing phase protects the copper matrix during friction. The ZrB_2_ particles resist the ceramic surface of the ball (Al_2_O_3_). Consequently, the presence of hard phases contributes to the composite’s increased hardness and wear resistance, which leads to a smaller loss of mass of the composite material.

## 4. Conclusions


Cu matrix composites reinforced with ZrB_2_ particles were successfully produced by Spark Plasma Sintering.Using ZrB_2_ as a reinforcing phase of the copper matrix improves its mechanical and tribological properties while decreasing its density, porosity, and corrosion resistance.Sintered copper without a reinforcing phase was characterized by the highest relative density (98%) and the lowest porosity (0.12). The relative density of the composites decreased from 96.0 to 90%, respectively, for 5–20% ZrB_2_ content.Composites with a higher ZrB_2_ content exhibit higher hardness, lower friction coefficient values, and specific wear rate values. The Cu-ZrB_2_ composites show very poor corrosion resistance in a sulphuric acid environment, probably due to the corrosion of the ceramic phase.


## Figures and Tables

**Figure 1 materials-17-06105-f001:**
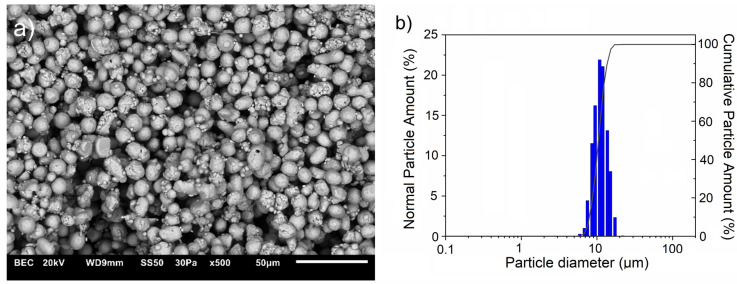
(**a**) SEM images of copper powder with (**b**) corresponding particle diameter distribution.

**Figure 2 materials-17-06105-f002:**
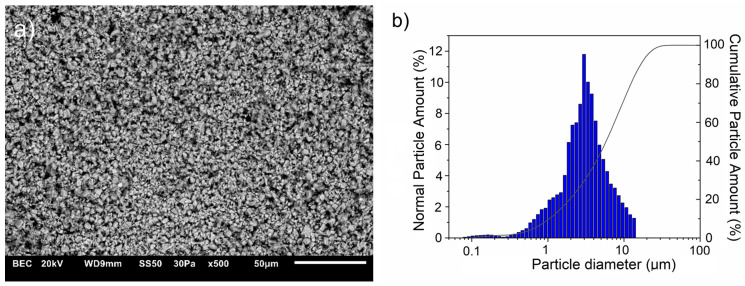
(**a**) SEM images of diboride zirconium powder with (**b**) corresponding particle diameter distribution.

**Figure 3 materials-17-06105-f003:**
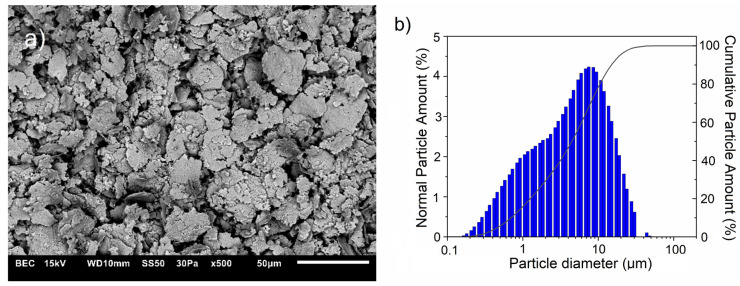
(**a**) SEM images with (**b**) corresponding particle diameter distribution of the powders with 20% ZrB_2_ after 10 h milling.

**Figure 4 materials-17-06105-f004:**
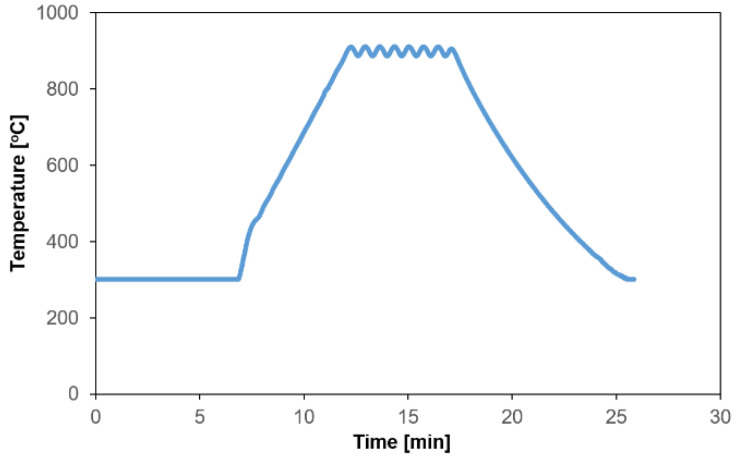
Heating curve of sintering Cu-5%ZrB_2_ composite.

**Figure 6 materials-17-06105-f006:**
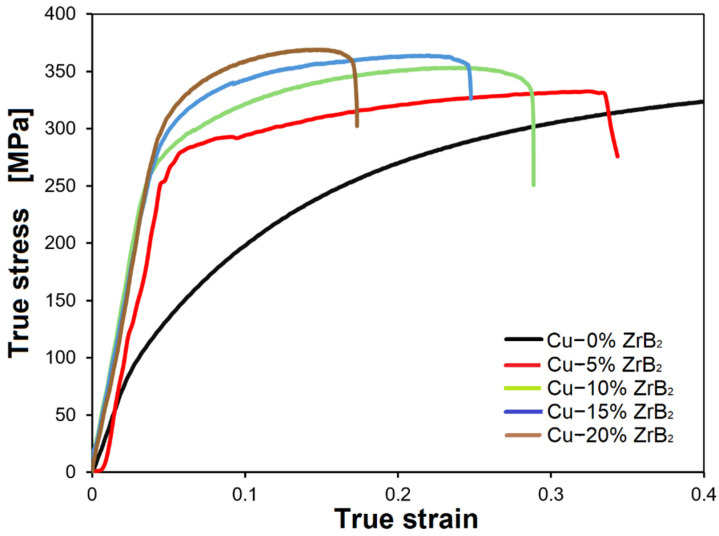
Compressive stress–strain curves of the sintered materials.

**Figure 7 materials-17-06105-f007:**
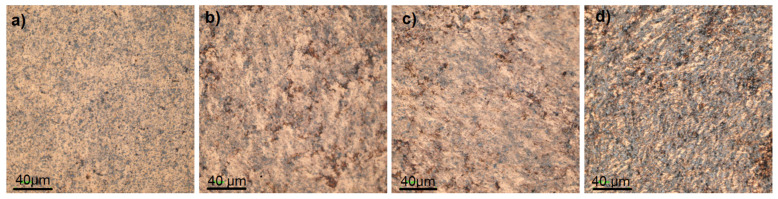
Microstructure of copper matrix composites with: (**a**) 5% ZrB_2_, (**b**) 10% ZrB_2_, (**c**) 15% ZrB_2_, and (**d**) 20% ZrB_2_ obtained from laser microscope.

**Figure 8 materials-17-06105-f008:**
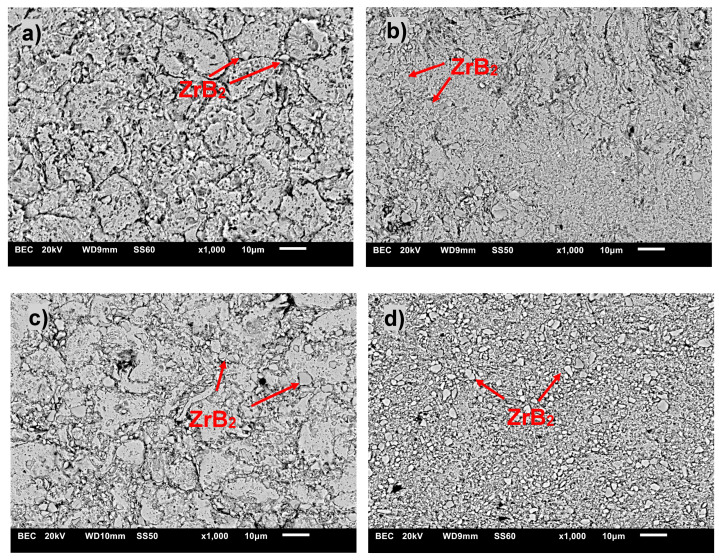
Microstructure (SEM) of copper matrix composites with: (**a**) 5% ZrB_2_, (**b**) 10% ZrB_2_, (**c**) 15% ZrB_2_, and (**d**) 20% ZrB_2_.

**Figure 9 materials-17-06105-f009:**
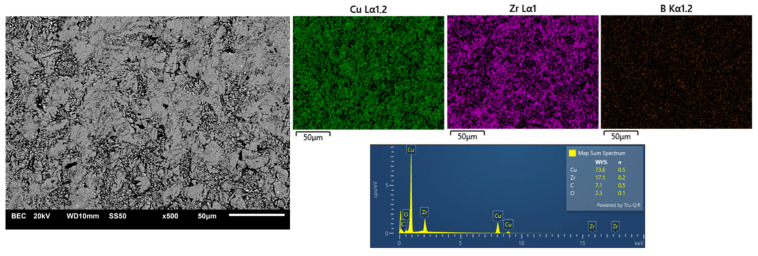
Selected microstructure (SEM) and EDS composition analysis of copper-20% ZrB_2_ composite.

**Figure 10 materials-17-06105-f010:**
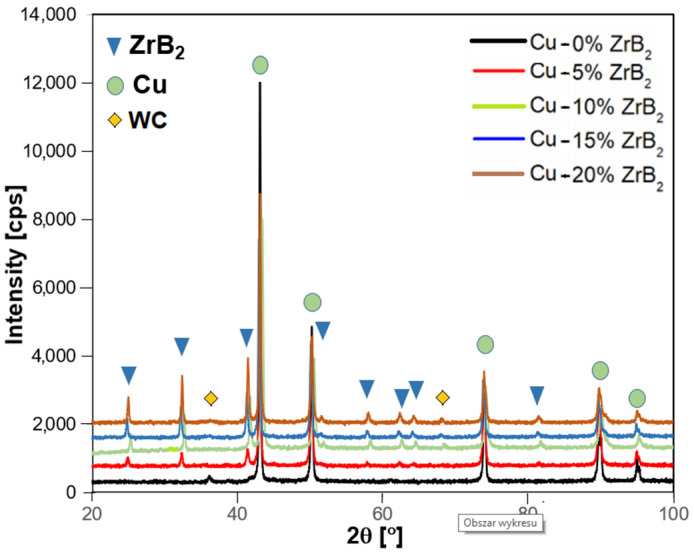
XRD patterns of copper matrix composites with different ZrB_2_ content.

**Figure 11 materials-17-06105-f011:**
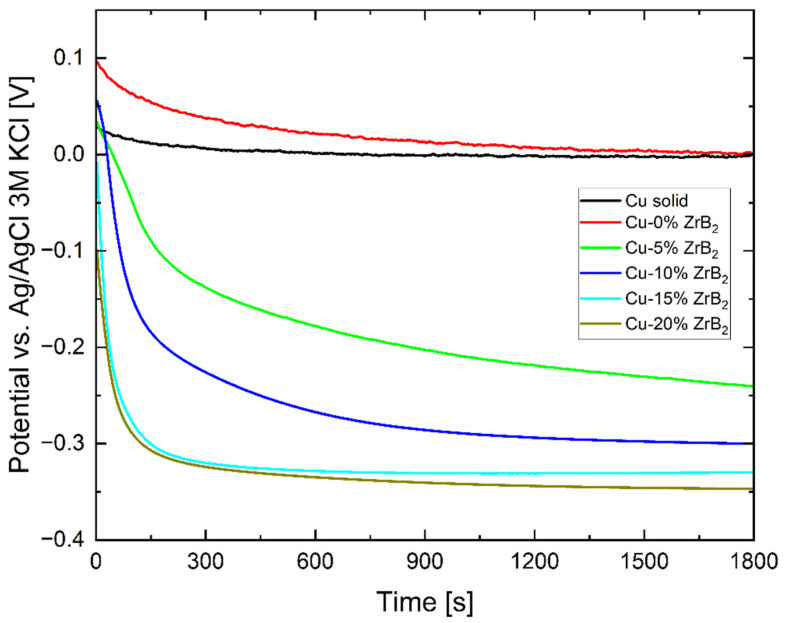
Open circuit potential recorded in 0.05 M H_2_SO_4_.

**Figure 12 materials-17-06105-f012:**
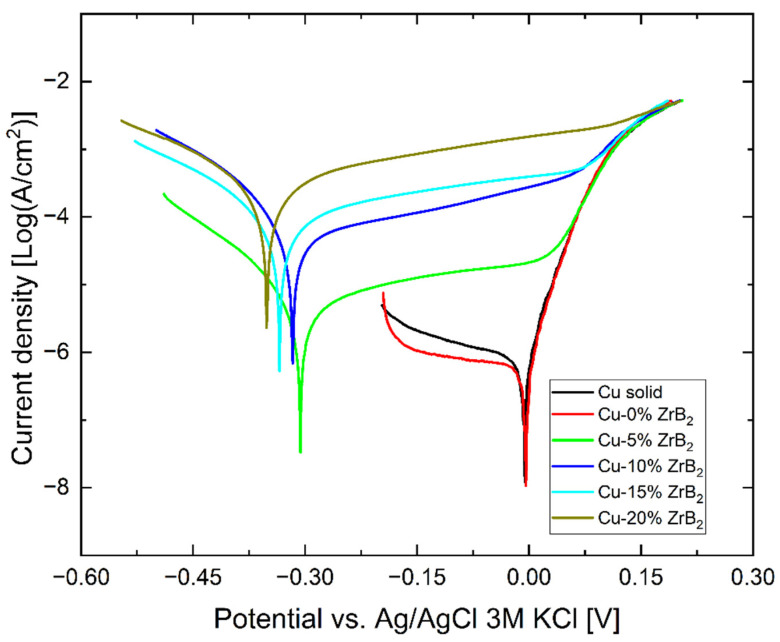
Polarization curves recorded on Cu and sintered Cu-ZrB_2_ composites obtained in deaerated 0.05 H_2_SO_4_ solution.

**Figure 13 materials-17-06105-f013:**
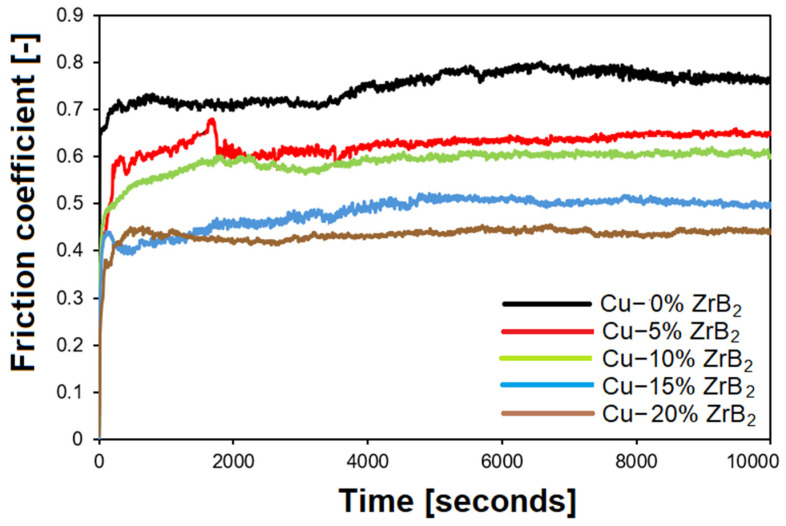
The friction coefficient of copper matrix composites with different ZrB_2_ content as a function of time test.

**Figure 14 materials-17-06105-f014:**
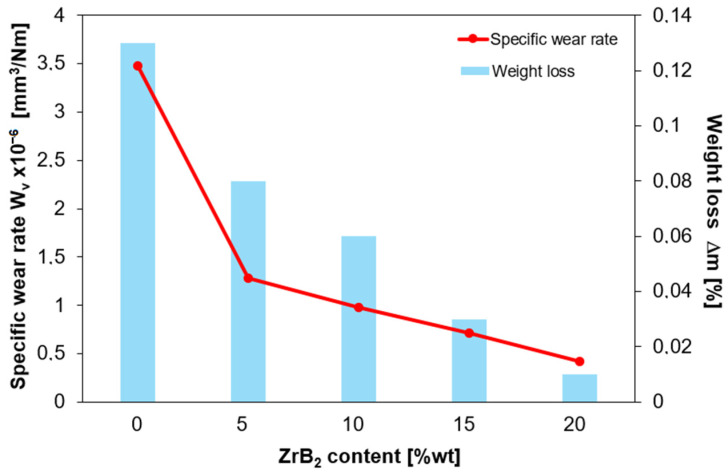
Variation of specific wear rate and weight loss as a function of ZrB_2_ content in the copper matrix.

**Figure 15 materials-17-06105-f015:**
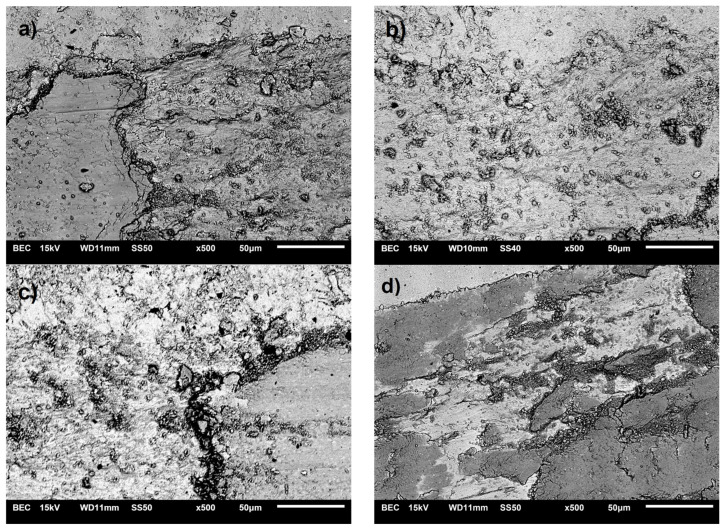
SEM micrograph of worn surface of copper matrix composites with: (**a**) 5% ZrB_2_, (**b**) 10% ZrB_2_, (**c**) 15% ZrB_2_, and (**d**) 20% ZrB_2_.

**Figure 16 materials-17-06105-f016:**
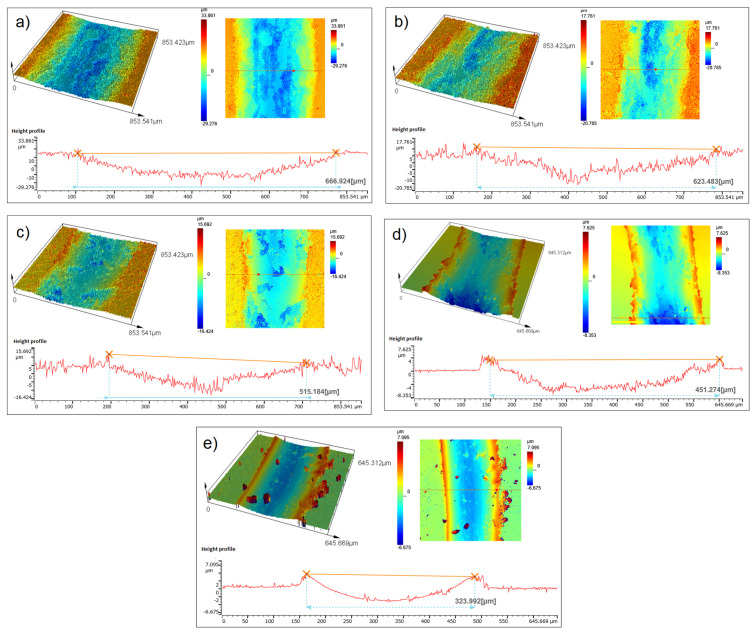
3D laser morphology of worn surface: (**a**) Cu-0% ZrB_2_, (**b**) Cu-5% ZrB_2_ composite, (**c**) Cu-10% ZrB_2_ composite, (**d**) Cu-15% ZrB_2_ composite and (**e**) Cu-20% ZrB_2_ composite.

**Figure 17 materials-17-06105-f017:**
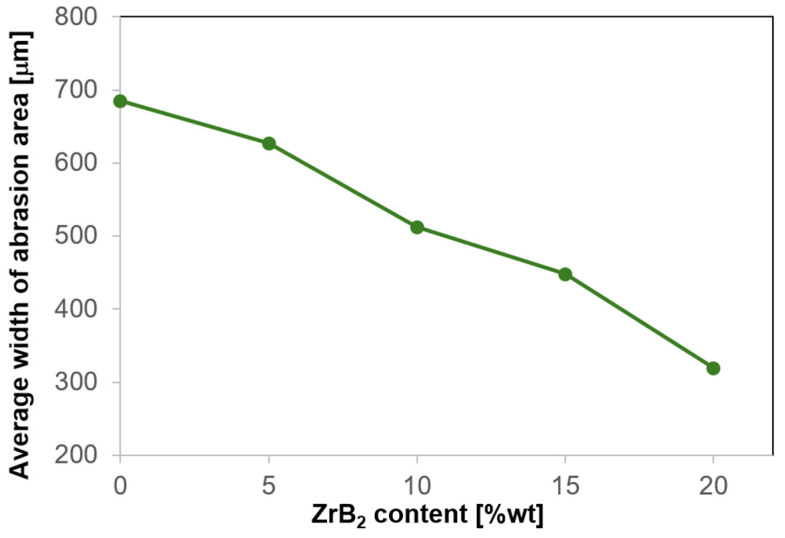
Variation of the average width of wear track as a function of ZrB_2_ content in the copper matrix.

**Table 2 materials-17-06105-t002:** Influence of the ZrB_2_ content on the properties of sintered materials.

Sintered Materials	Apparent Density [g/cm^3^]	Relative Density [%]	Open Porosity [%]	Young’s Modulus [GPa]	Hardness HV0.3 [-]
Cu-0% ZrB_2_	8.72	98	0.12	107	61.2
Cu-5% ZrB_2_	8.41	96	1.91	120	105
Cu-10% ZrB_2_	8.06	94	3.45	131	133
Cu-15% ZrB_2_	7.59	91	6.86	141	140
Cu-20% ZrB_2_	7.34	90	8.85	148	161

**Table 3 materials-17-06105-t003:** Potentiodynamic polarization parameters for copper and Cu-ZrB_2_ composites in 0.05 M H_2_SO_4_ solution.

Sample	OCP [mV]	E_corr_ [mV]	I_corr_ µA/cm^2^	β-Anodic Tafel Const.	β Cathodic Tafel Const.
Cu solid	4 ± 4	−3 ± 5	0.7 ± 0.1	33 ± 1	−361 ± 20
Cu-0% ZrB_2_	1 ± 3	−8 ± 3	0.6 ± 0.05	31 ± 1	−610 ± 27
Cu-5% ZrB_2_	285 ± 20	302 ± 9	7.3 ± 2	595 ± 20	−120 ± 15
Cu-10% ZrB_2_	290 ± 7	315 ± 8	58 ± 7	509 ± 48	−107 ± 9
Cu-15% ZrB_2_	340 ± 10	345 ± 10	133 ± 20	650 ± 70	−178 ± 5
Cu-20% ZrB_2_	340 ± 15	350 ± 12	289 ± 27	467 ± 20	−193 ± 8

## Data Availability

The original contributions presented in this study are included in the article. Further inquiries can be directed to the corresponding author.

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
