# Peer review of "Effect of ZrB_2_ Content on the Properties of Copper Matrix Composite"

_materials, 2024, doi:10.3390/ma17246105_

Round 1

Reviewer 1 Report

Comments and Suggestions for Authors

The manuscript «Effect of ZrB2 Content on the Properties of Copper Matrix Composite» presents a study on modifying ductile metal matrices with hard ceramic particles. This article couldbe placed on a par with other works devoted to improving copper-based metal matrix composite materials. However, there are some shortcomings that need to be solved before publication. Here are the main comments.

1. In Figure 2, the histogram of the particle size distribution does not correlate well with the powder image. The powder appears finer in the image. Please check if there is an error here.

2. The histogram in Figure 3b almost completely repeats the histogram in Figure 2b. Perhaps this is a technical error in the layout of the article? In addition, despite the obvious similarity of the particle size distribution, the differences in the particle size are quite obvious during microscopic examination (Figures 2a and 3a).

3. Why 0.05 M H2SO4 solution was chosen for corrosion test? Is it related to potential areas of application of the composite?

4. Table 1. How was open porosity measured in the work?

5. Figure 5. The scale bars are barely visible.

6. Figure 8. Please mark the phases in the SEM images of Figures 8 and 9. According to the histograms of the particle size distribution, more than 50% of the ZrB2 particles are larger than 3 µm and should be clearly visible in the microstructure.

7. Figure 8, EDX spectre. Concentration of Zr is higher than it should be for this composition. Was this spectre made from the area with higher ZrB2 concentration?

8. Figure 10. XRD patterns contain some unidentified peaks. If it is a contamination from WC grinding media, it could be mentioned in the Figure 10 or in supporting text of the article.

9. Line 286. A misprint – ZrB2.

10. Introduction, Lines 31-35. Perhaps the lack of mutual solubility between copper and zirconium boride should be also mentioned here.

Author Response

Dear Reviewer,

Thank you for your detailed and insightful comments on our manuscript. We greatly appreciate the time and effort you invested in reviewing our work. Your feedback has been very helpful in refining the manuscript. We have carefully reviewed your suggestions and incorporated the necessary revisions to address your concerns. Please find our point-by-point responses to your comments below.

Reviewer 1

The manuscript «Effect of ZrB2 Content on the Properties of Copper Matrix Composite» presents a study on modifying ductile metal matrices with hard ceramic particles. This article could be placed on par with other works on improving copper-based metal matrix composite materials. However, some shortcomings need to be solved before publication. Here are the main comments.

Question 1: In Figure 2, the histogram of the particle size distribution does not correlate well with the powder image. The powder appears finer in the image. Please check if there is an error here.

Answer 1:   The correct result of the particle size distribution has been inserted in Figure 2 particle size distribution.

Question 2: The histogram in Figure 3b almost wholly repeats the histogram in Figure 2b. Perhaps this is a technical error in the layout of the article. In addition, despite the apparent similarity of the particle size distribution, the differences in the particle size are pretty evident during microscopic examination (Figures 2a and 3a).

Answer 2:   Thank you for your attention. This is our technical error. Figures 2 and 3 mistakenly included the same grain distribution analyses. Figure 2b now includes the correct result of the particle size distribution.

Question 3: Why was 0.05 M H2SO4 solution chosen for the corrosion test? Is it related to potential areas of application of the composite?

Answer 3:   Sulfuric acid is widely used as an electrolyte for corrosion testing due to its good electrical conductivity and widespread use in industry. Many publications make it possible to compare the results obtained with those of other authors. Moreover, many lubricants contain sulfur-containing compounds as anti-wear additives, which, upon exposure to oxidizing atmospheres, elevated temperatures, and moisture, can degrade with the formation of sulfuric acid. Hence, knowing the corrosion mechanism of such materials in sulfuric acid solutions seems extremely important.

Question 4: Table 1. How was open porosity measured in the work?

Answer 4:   The hydrostatic (Archimedes) method determined apparent density and open porosity. The measurement consists of determining the mass of a dry sample in air (ms), saturating the sample with distilled water and determining the mass of the sample during weighing in air (mn), hydrostatic weighing (in water) of the water-saturated sample (mw). Density [g/m3] was calculated according to the formula:

                   Open porosity (Po) [%] was calculated according to the formula:

                   where:

                   ms- dry sample weight [g]

                   mn – mass of a sample saturated with liquid, weighed in distilled water [g]

                   mw – mass of the sample saturated with liquid, weighed in air [g]

                   rL- density of distilled water [g/m3].

Question 5: Figure 5. The scale bars are barely visible.

Answer 5:   It has been corrected.

Question 6: Figure 8. Please mark the phases in the SEM images of Figures 8 and 9. According to the particle size distribution histograms, more than 50% of the ZrB2 particles are larger than 3 µm and should be clearly visible in the microstructure.

Answer 6:   Figure 8 has been changed. Now, it compares microstructures (at 1000x magnification) of composites containing 5, 10, 15 and 20 wt.% ZrB2. Additionally, the reinforcing phase ZrB2 is marked on the microstructures.

Question 7  Figure 8, EDX spectre. The concentration of Zr is higher than it should be for this composition. Was this spectre made from the area with higher ZrB2 concentration?

Answer 7:   This article presents the results of surface EDS analysis. The area with higher ZrB2 concentration was not explicitly selected. The discrepancy between the actual composition and the measured ZrB2 concentration is due to the specifics of the EDS analysis. It should be remembered that the data used for EDS analysis are not from the cross-sectional plane but rather from a specific volume on the sample surface (the region of electron beam excitation). If the ZrB2 phase was located just below the studied surface, it was also detected and included in the presented results.

Question 8: Figure 10. XRD patterns contain some unidentified peaks. If it is a contamination from WC grinding media, it could be mentioned in Figure 10 or the supporting text of the article.

Answer 8:   Thank you for your valuable attention. The milling process was carried out using a Fritsch Pulveristte 5 planetary ball mill, which was a container with balls and WC. A side effect of the milling is the introduction of WC contamination into the powder mixture. The XRD patterns (Fig. 10) have been corrected. Unidentified peaks come from WC.

Question 9: Line 286. A misprint – ZrB2.

Answer 9:   It has been corrected.

Question 10:   Introduction, Lines 31-35. Perhaps the lack of mutual solubility between copper and zirconium boride should also be mentioned here.

Answer 10:      Since the melting point of the phases involved is not exceeded, we do not need to consider the phenomenon of mutual solubility of the two components in the system we are studying.

Reviewer 2 Report

Comments and Suggestions for Authors

This manuscript studies the mechanical and tribological properties of Cu matrix composites reinforced with ZrB2 particles, which were produced by Spark Plasma Sintering. Composites with a higher ZrB2 content exhibit higher hardness, lower friction coefficient values and specific wear rate values. However, the Cu- ZrB2 composites show very poor corrosion resistance in a sulphuric acid environment, probably due to the corrosion of the ceramic phase. This is a routine material test experimental report. This kind of materials have been studied on other papers [reference 41]. Therefore, the novelty is not very high in this manuscript. However, the manuscript is acceptable for this new test data reported. Nevertheless, it is better to have a comparison table for Cu with different insertion ceramic materials, so that it will be of more valuable for the material scientists and engineers to select which inclusions is better for the real applications.   

There are still typos that needed to be corrected:

In Equation (2) and (3) the units of E0 are missed.\

Line 212, in the Equation (3), the -2210 should be -0.210 eV.

Line 216: What is without reaction (II), and direction of reaction (2)?

Line 221: (0.25 M Na2SO4 + 0.05 M H2SO4, pH=1), all the 2 and 4 should be in subscripts.

Line 249: ZrB2, the 2 should be in subscript.

Line 254:  ZrCl6, the 6 should be in subscript.

Author Response

Dear Reviewer,

Thank you for your detailed and insightful comments on our manuscript. We greatly appreciate the time and effort you invested in reviewing our work. Your feedback has been very helpful in refining the manuscript. We have carefully reviewed your suggestions and incorporated the necessary revisions to address your concerns. Please find our point-by-point responses to your comments below

Reviewer 2

            This manuscript studies the mechanical and tribological properties of Cu matrix composites reinforced with ZrB2 particles, which Spark Plasma Sintering produced. Composites with a higher ZrB2 content exhibit higher hardness, lower friction coefficient values and specific wear rate values. However, the Cu- ZrB2 composites show very poor corrosion resistance in a sulphuric acid environment, probably due to the corrosion of the ceramic phase. This is a routine material test experimental report. This kind of material has been studied in other papers [reference 41]. Therefore, the novelty of this manuscript is not very high.

Question 1: However, the manuscript is acceptable for the reported new test data. Nevertheless, having a comparison table for Cu with different insertion ceramic materials is better. It will be more valuable for the material scientists and engineers to select which inclusions are better for real applications.

Answer 1:   Thank you very much for your attention. The theoretical part now includes an additional table 1 with literature data for composites reinforced with various ceramic materials.

Question 2: There are still typos that need to be corrected:

                   In Equation (2) and (3) the units of E0 are missed.\

                   Line 212, in the Equation (3), the "-2210" should be "-0.210 eV".

                   Line 216: What is " without reaction (II)", and "direction of reaction (2)"?

                   Line 221: (0.25 M Na2SO4 + 0.05 M H2SO4, pH=1), all the "2" and "4" should be in subscripts.

                   Line 249: ZrB2, the "2" should be in subscript.

                   Line 254:  ZrCl6, the "6" should be in subscript.

Answer 2:   All have been corrected.

Reviewer 3 Report

Comments and Suggestions for Authors

2. Materials and Methods

Are presented results the same as presented in

I8. Sulima, G. Boczkal, Processing and Properties of ZrB2-Copper Matrix Composites Produced by Ball Milling and Spark Plasma Sintering, Materials 16 (2023), 7455

if this is the cas, please do not duplilcate the results, but part of the used materials and their characterization scould be cited.

The corrosion test was carried out using a 0.05 M H2SO4 solution. Why did zou chose these conditons for corrosion exsamination?

Figures 8 and 9 are given for 5 and 20 % of ZrB2 composite. Please explain why did you not present the results for other contents, as other results are presented for serias fro 5 to 20 % ZrB2 content ( Figs .6,10,11, 12, 13, 14. )

Fig2 16 and 17. present the SEM of 10 and 20 % of ZrB2 composite. Please use figs for all or selected samples, but use the same content of of ZrB2  for all selected figs.

Author Response

Dear Reviewers,

Thank you for your detailed and insightful comments on our manuscript. We greatly appreciate the time and effort you invested in reviewing our work. Your feedback has been very helpful in refining the manuscript. We have carefully reviewed your suggestions and incorporated the necessary revisions to address your concerns. Please find our point-by-point responses to your comments below.

Reviewer 1

The manuscript «Effect of ZrB2 Content on the Properties of Copper Matrix Composite» presents a study on modifying ductile metal matrices with hard ceramic particles. This article could be placed on par with other works on improving copper-based metal matrix composite materials. However, some shortcomings need to be solved before publication. Here are the main comments.

Question 1: In Figure 2, the histogram of the particle size distribution does not correlate well with the powder image. The powder appears finer in the image. Please check if there is an error here.

Answer 1:   The correct result of the particle size distribution has been inserted in Figure 2 particle size distribution.

Question 2: The histogram in Figure 3b almost wholly repeats the histogram in Figure 2b. Perhaps this is a technical error in the layout of the article. In addition, despite the apparent similarity of the particle size distribution, the differences in the particle size are pretty evident during microscopic examination (Figures 2a and 3a).

Answer 2:   Thank you for your attention. This is our technical error. Figures 2 and 3 mistakenly included the same grain distribution analyses. Figure 2b now includes the correct result of the particle size distribution.

Question 3: Why was 0.05 M H2SO4 solution chosen for the corrosion test? Is it related to potential areas of application of the composite?

Answer 3:   Sulfuric acid is widely used as an electrolyte for corrosion testing due to its good electrical conductivity and widespread use in industry. Many publications make it possible to compare the results obtained with those of other authors. Moreover, many lubricants contain sulfur-containing compounds as anti-wear additives, which, upon exposure to oxidizing atmospheres, elevated temperatures, and moisture, can degrade with the formation of sulfuric acid. Hence, knowing the corrosion mechanism of such materials in sulfuric acid solutions seems extremely important.

Question 4: Table 1. How was open porosity measured in the work?

Answer 4:   The hydrostatic (Archimedes) method determined apparent density and open porosity. The measurement consists of determining the mass of a dry sample in air (ms), saturating the sample with distilled water and determining the mass of the sample during weighing in air (mn), hydrostatic weighing (in water) of the water-saturated sample (mw). Density [g/m3] was calculated according to the formula:

                   Open porosity (Po) [%] was calculated according to the formula:

                   where:

                   ms- dry sample weight [g]

                   mn – mass of a sample saturated with liquid, weighed in distilled water [g]

                   mw – mass of the sample saturated with liquid, weighed in air [g]

                   rL- density of distilled water [g/m3].

Question 5: Figure 5. The scale bars are barely visible.

Answer 5:   It has been corrected.

Question 6: Figure 8. Please mark the phases in the SEM images of Figures 8 and 9. According to the particle size distribution histograms, more than 50% of the ZrB2 particles are larger than 3 µm and should be clearly visible in the microstructure.

Answer 6:   Figure 8 has been changed. Now, it compares microstructures (at 1000x magnification) of composites containing 5, 10, 15 and 20 wt.% ZrB2. Additionally, the reinforcing phase ZrB2 is marked on the microstructures.

Question 7  Figure 8, EDX spectre. The concentration of Zr is higher than it should be for this composition. Was this spectre made from the area with higher ZrB2 concentration?

Answer 7:   This article presents the results of surface EDS analysis. The area with higher ZrB2 concentration was not explicitly selected. The discrepancy between the actual composition and the measured ZrB2 concentration is due to the specifics of the EDS analysis. It should be remembered that the data used for EDS analysis are not from the cross-sectional plane but rather from a specific volume on the sample surface (the region of electron beam excitation). If the ZrB2 phase was located just below the studied surface, it was also detected and included in the presented results.

Question 8: Figure 10. XRD patterns contain some unidentified peaks. If it is a contamination from WC grinding media, it could be mentioned in Figure 10 or the supporting text of the article.

Answer 8:   Thank you for your valuable attention. The milling process was carried out using a Fritsch Pulveristte 5 planetary ball mill, which was a container with balls and WC. A side effect of the milling is the introduction of WC contamination into the powder mixture. The XRD patterns (Fig. 10) have been corrected. Unidentified peaks come from WC.

Question 9: Line 286. A misprint – ZrB2.

Answer 9:   It has been corrected.

Question 10:   Introduction, Lines 31-35. Perhaps the lack of mutual solubility between copper and zirconium boride should also be mentioned here.

Answer 10:      Since the melting point of the phases involved is not exceeded, we do not need to consider the phenomenon of mutual solubility of the two components in the system we are studying.

Reviewer 2

            This manuscript studies the mechanical and tribological properties of Cu matrix composites reinforced with ZrB2 particles, which Spark Plasma Sintering produced. Composites with a higher ZrB2 content exhibit higher hardness, lower friction coefficient values and specific wear rate values. However, the Cu- ZrB2 composites show very poor corrosion resistance in a sulphuric acid environment, probably due to the corrosion of the ceramic phase. This is a routine material test experimental report. This kind of material has been studied in other papers [reference 41]. Therefore, the novelty of this manuscript is not very high.

Question 1: However, the manuscript is acceptable for the reported new test data. Nevertheless, having a comparison table for Cu with different insertion ceramic materials is better. It will be more valuable for the material scientists and engineers to select which inclusions are better for real applications.

Answer 1:   Thank you very much for your attention. The theoretical part now includes an additional table 1 with literature data for composites reinforced with various ceramic materials.

Question 2: There are still typos that need to be corrected:

                   In Equation (2) and (3) the units of E0 are missed.\

                   Line 212, in the Equation (3), the "-2210" should be "-0.210 eV".

                   Line 216: What is " without reaction (II)", and "direction of reaction (2)"?

                   Line 221: (0.25 M Na2SO4 + 0.05 M H2SO4, pH=1), all the "2" and "4" should be in subscripts.

                   Line 249: ZrB2, the "2" should be in subscript.

                   Line 254:  ZrCl6, the "6" should be in subscript.

Answer 2:   All have been corrected.

Reviewer 3

Question 1: Are the presented results the same as presented in I8. Sulima, G. Boczkal, Processing and Properties of ZrB2-Copper Matrix Composites Produced by Ball Milling and Spark Plasma Sintering, Materials 16 (2023), 7455 if this is the cas, please do not duplicate the results, but part of the used materials and their characterization should be cited.

Answer 1:   The results presented in the article are not the same as those in the publication: I. Sulima, G. Boczkal, Processing and Properties of ZrB2-Copper Matrix Composites Produced by Ball Milling and Spark Plasma Sintering, Materials 16 (2023), 7455. For this article, composite materials (Cu-xZrB2 composites (x = 5, 10, 15, 20 wt.%) were sintered by the SPS method at a shorter time of 3 min at 900°C. The results of measurements of density, open porosity, Young's modulus and hardness for these composite materials are different.

Question 2: The corrosion test was carried out using a 0.05 M H2SO4 solution. Why did zou chose these conditions for corrosion examination?

Answer 2:   Sulfuric acid is widely used as an electrolyte for corrosion testing due to its good electrical conductivity and widespread use in industry. Many publications make it possible to compare the results obtained with those of other authors. Moreover, many lubricants contain sulfur-containing compounds as anti-wear additives, which, upon exposure to oxidizing atmospheres, elevated temperatures, and moisture, can degrade with the formation of sulfuric acid. Hence, knowing the corrosion mechanism of such materials in sulfuric acid solutions seems extremely important.

Question 3: Figures 8 and 9 are given for 5 and 20 % of ZrB2 composite. Please explain why did you not present the results for other contents, as other results are presented for series from 5 to 20 % ZrB2 content ( Figs .6,10,11, 12, 13, 14. )

Answer 3:   The first version of the article included sample microstructures for selected composites. Figure 8 has been changed. Now it compares microstructures (at 1000x magnification) of composites containing 5, 10, 15 and 20 wt.% ZrB2. Additionally, the reinforcing phase ZrB2 is marked on the microstructures. Figure 9 presents an example result of EDS analysis for a composite containing 20% ZrB2. In the case of EDS analyses, the results for the other sintered composites were very similar. Therefore, the article limited the presentation of results to one selected composite sinter.

Question 4: Fig2 16 and 17. present the SEM of 10 and 20 % of ZrB2 composite. Please use figs for all or selected samples, but use the same content of of ZrB2  for all selected figs.

Answer 4:   It has been corrected. Figure 15 shows the micrographs of the worn surface of copper matrix composites with  5-20 wt%.

Round 2

Reviewer 1 Report

Comments and Suggestions for Authors

Thank you for responding to the comments.

Reviewer 3 Report

Comments and Suggestions for Authors

After corrections,  in this state the manuscript is appropriate for publishing.